# Glycidyl and Methyl Methacrylate UV-Grafted PDMS Membrane Modification toward Tramadol Membrane Selectivity

**DOI:** 10.3390/membranes11100752

**Published:** 2021-09-30

**Authors:** Mahdi Bourassi, Mariia Pasichnyk, Oscar Oesch, Swati Sundararajan, Tereza Trávničková, Karel Soukup, Roni Kasher, Jana Gaálová

**Affiliations:** 1Faculty of Science Institute for Environmental Studies, Charles University, Benátská 2, 128 01 Prague, Czech Republic; 2Institute of Chemical Process Fundamentals of the ASCR, v.v.i. Rozvojova 135, 165 00 Prague, Czech Republic; pasichnyk@icpf.cas.cz (M.P.); travnickovat@icpf.cas.cz (T.T.); soukup@icpf.cas.cz (K.S.); gaalova@icpf.cas.cz (J.G.); 3Institut de Chimie des Milieux et Matériaux de Poitiers, Poitiers University, 86073 Poitiers, France; oscaroesch@gmail.com; 4Department of Desalination & Water Treatment, Zuckerberg Institute for Water Research, Jacob Blaustein Institutes for Desert Research, Ben-Gurion University of the Negev, Sede Boqer Campus, Midreshet Ben-Gurion 8499000, Israel; swati14.ssr@gmail.com (S.S.); kasher@bgu.ac.il (R.K.)

**Keywords:** membrane separation, pharmaceutical’s pertraction, selective membrane, water depollution, UV-grafting modification

## Abstract

Pharmaceutical wastewater pollution has reached an alarming stage, as many studies have reported. Membrane separation has shown great performance in wastewater treatment, but there are some drawbacks and undesired byproducts of this process. Selective membranes could be used for pollutant investigation sensors or even for pollutant recovery. The polydimethylsiloxane (PDMS) membrane was first tested on separated and mixed antibiotic (ATB) water solutions containing sulfamethoxazole (SM), trimethoprim (TMP), and tetracycline (TET). Then, the bare and ultra-violet grafted (UV-grafted) PDMS membranes (MMA-DMAEMA 10, GMA-DMAEMA 5, and GMA-DMAEMA 10) were tested in tramadol (TRA) separation, where the diffusion coefficient was evaluated. Finally, the membranes were tested in pertraction with a mixture of SM, TMP, TET, and TRA. The membranes were characterized using the following methods: contact angle measurement, FTIR, SEM/EDX, and surface and pore analysis. The main findings were that TET was co-eluted during mixed ATB pertraction, and GMA-DMAEMA 5 was found to selectively permeate TRA over the present ATBs.

## 1. Introduction

Pharmaceutical wastewater has become alarming lately. These pollutants originate from different domains such as pharmaceutical industries, which are the producers, then the consumers, such as households, hospitals, livestock, aquaculture, and agriculture. Most of these producers and consumers are connected to conventional wastewater treatment plants. However, most of these pollutants persist after conventional treatment and flow into the environment. For example, Diaz-Sosa et al. detected sulfamethoxazole, tramadol, and trimethoprim, with concentrations of 530, 870, and 500 ng/L, respectively, as secondary effluents of a wastewater treatment plant [1]. This effluent channel was directly discharged to a river. Many studies have reported the presence of different emerging contaminants in rivers, lakes, wells, and even in soils and organisms at different sampling sites around the globe [2,3,4,5].

The presence of one drug in the environment, even at trace concentrations, can be harmful for living organisms [4,5,6]. Even more, with the presence of different pharmaceuticals, the synergistic effect can be much more serious [7]. For proper treatment of this kind of pollutant, advanced treatment should be applied directly to the source. The most pertinent advanced treatments are adsorptions, oxidations, and membrane separation. Most of these technologies have advantages and drawbacks. For the adsorbents, they perform well in terms of pollutant elimination [8,9]. Thus, these adsorbents are limited by the adsorption capacity, and they need to be regenerated or incinerated. These two processes are hardly handled [10]. Advanced oxidation processes are efficient for highly concentrated wastewaters. However, they are energy-demanding and need to degrade pollutants properly, avoiding the regeneration of by-products, which are usually harmful to the environment [11]. Membrane separation processes are widely used due to their high separation competence and varied adjustable parameters, even though most of these processes are pressure-driven. However, they undergo fouling and clogging, where the membrane should be changed, and after separation, they generate concentrated solutions, which are hard to manage [12].

Membrane separation processes are broadly applied at full scale. This large interest in membrane separation processes, regardless of their disadvantages, is due to the extensive physical and chemical changeable parameters. The physical parameters of the process are temperature, solution flow, pressure, and stirring system [12]. The chemical composition of the membrane has the main effect on the process, an example being membrane-based carbon nanotubes which combine the high adsorption of carbon nanotubes and membrane process [13]. The composited membrane or supported membrane, such as polydimethylsiloxane (PDMS) supported on polyethylene terephthalate (PET), which is commercially available, shows interesting separation factors, grate stability, and moderate selectivity [14,15]. PDMS and membrane surface modification technologies aim to improve hydrophobicity, molecular, or even chiral selectivity [16]. PDMS membranes are more often used for pervaporation and the separation of volatile compounds, thanks to its non-porous structure [17,18]. Thus, recent studies have focused on liquid–liquid selective separations [19,20].

The membrane surface is a key aspect for membrane selectivity. The separation process begins with adsorption of the pollutant on the membrane surface, then diffusion through the membrane, then desorption in the permeate side. The surface should be adapted to targeted pollutant, whether to separate the pollutant by attraction or repulsion. While seeking membrane selectivity, different membrane surface modifications are available as the addition of a chemical modifier such as sulfonation, carboxylation, epoxidation, and amination, or chemical grafting [19,20,21]. Chemical graft polymerization was shown as a facile method to alter membrane surface properties due to the choice of monomers with diverse sidechain groups [22,23,24].

This study focuses on the examination of different membrane pollutant interactions in separate and mixed solutions. Much more attention is oriented to developing a selective membrane, by modifying the membrane surface using UV-grafting polymerization. The results of this study promise interesting applications of this selective membrane, including specific pollutant supervision using sensors, or even separation for pollutant recovery.

## 2. Materials and Methods

### 2.1. Materials

For this study, different materials were used. The polydimethylsiloxane PERVAP 4060 (PDMS) membrane purchased from DeltaMem AG. The nonporous membrane is widely available and had shown interesting separation performances [17,18]. Tramadol (TRA), sulfamethoxazole (SM), trimethoprim (TMP), and tetracycline (TET), with high purity of ≥99%, ≥98%, ≥98%, and 98%, respectively, were used for model solutions preparation. All four of these pharmaceuticals were HPLC grade and were ordered from Sigma-Aldrich. For the membrane modification agents, methyl methacrylate (MMA), glycidyl methacrylate (GMA), 2-(dimethylamino)ethyl methacrylate (DMAEMA), benzophenone, ethanol, and dioxane were used. Last, the ultrapure water used was collected from ultrapure water system Simple Lab, Millipore S. A., Molsheim, France, with of 18.2 MΩ cm resistivity at 25 °C.

### 2.2. Membranes Modification

The PDMS membranes were modified using ultra-violet (UV)-initiated graft polymerization as previously published with slight changes [21]. Briefly, the monomers used for the modification were either GMA or MMA with DMAEMA in a 9:1 mole ratio. A total monomer concentration of 0.8 M was maintained for all modifications. The modifications were carried out on the active side of the PDMS membrane following the steps schematized in Figure 1. Round coupons of PDMS membranes (50 mm diameter) were immersed in water prior to the modification, followed by washing of the membrane with a water–ethanol mixture (1:4 ratio) for 1 min. Meanwhile, benzophenone (0.05 M) solution was prepared in a 1:4 ratio of water and ethanol mixture, and the monomer solution was prepared in dioxane. The membranes were mounted in a record exposing only the active side, and the prepared benzophenone solution was added to the membrane for 1 min. Afterwards, the membranes were washed again with a 1:4 ratio of water and ethanol mixture for 1 min, followed by pre-incubation of the membranes with the monomer solution in dioxane for 1 min. Then, the record was placed in a UV chamber (Intelli-Ray 400, UV-tron International, MA, USA) for grafting under UV irradiation (mercury lamp; intensity of 33.056 mW/cm^2^ as measured by a UV light meter) for a period of either 5 or 10 min, depending on the reaction conditions. The modified surface was washed using dioxane and a water–ethanol mixture (1:4 ratio) and was sonicated 3 times with distilled water for 10 min each and stored in double distilled water at 4 °C until further use.

### 2.3. Characterization Methods

The characterization of both membranes and solutions during this study helped us to interpret and discuss the results. The liquid samples were analyzed using high-performance liquid chromatography coupled with ultra-violet detector (HPLC-UV). Concerning membrane characterization, contact angle, scanning electron microscope (SEM) imaging, Fourier transform infrared (FTIR) spectra, specific surface characterizations, and energy-dispersive X-ray spectroscopy (EDX) were analyzed.

#### 2.3.1. HPLC-UV Analysis

We used an 1100 Series HPLC from Agilent Technologies instrument equipped with a binary pump, degasser, for liquid samples analysis. Pollutant separation was achieved by using the chromatographic column Luna C18 (5 µm, 4.6 × 150 mm, Phenomenex). The auto sampler injected 20 µL of sample. The injections were eluted with water, acetonitrile, and methanol (80:10:10) and acidified with formic acid (0.1% vol) at a 0.3 mL/min flow rate at 20 °C. The separated pollutants were detected at a 254 nm wavelength with a diode array detector (DAD).

#### 2.3.2. Contact Angle Measurements

Water drop contact angle measurements were carried out using OCA 20 Dataphysics Products, Filderstadt, Germany. A volume of 2 µL was dropped on the surface of membrane. The images collected from the membrane–water drop contact were treated using ImageJ software. The angle measurement value indicates an average of three readings.

#### 2.3.3. SEM Imaging and EDX Analysis

We investigated the morphology of the tested membranes using SEM with a field emission gun (FEG) electron source Tescan Lyra dual beam microscope. The elemental composition of the modified membranes was analyzed using an EDX analyzer and detected by XMaxN with a 20 mm^2^ Silicon drift detector (SDD) (Oxford Instruments). AZtecEnergy software monitored the measurements and gathered results. The samples were placed on conductive carbon tape, and samples analysis was carried out using 15 kV electron beam analysis.

#### 2.3.4. FTIR Spectra

The tested membranes were analyzed by FTIR with an Avatar 360 Nicolet spectrometer. FTIR measurements ranged from 508 cm^−1^ to 4000 cm^−1^. Each membrane was scanned 200 times with resolution of 1.93 cm^−1^. The analyses were achieved in attenuated total reflection (ATR) mode where membranes were pressed against the ZnSe crystal to be measured.

#### 2.3.5. Specific Surface Characterization

Using the automated volumetric gas adsorption analyzer ASAP 2020 (Micromeritics, Norcross, GA), nitrogen physisorption was measured on the tested membranes at 77 K. To guarantee the accuracy of the obtained adsorption isotherms, highly pure nitrogen (99.999 vol. %) as well as helium (99.996 vol. %) were used for the analysis. All samples were degassed at 70 °C under vacuum for 12 h before nitrogen adsorption measurements.

### 2.4. Preparation of Model Solutions

To test all membranes, different solution loads with the model pollutants SM, TMP, TET, and TRA were prepared (see Table 1). First, separated antibiotics solutions of sulfamethoxazole, trimethoprim, and tetracycline separately with the concentration of 200 mg/L of each antibiotic were made. A mixture of these antibiotics was prepared with 100 mg/L of each antibiotic SM, TMP, and TET altogether. The prepared antibiotics solutions were tested on bear PDMS membranes. However, a prepared tramadol solution with a concentration of 500 mg/L was tested on all membranes (bare PDMS, MMA-DMAEMA 10, GMA-DMAEMA 10, and GMA-DMAEMA 5). Finally, all three antibiotics and TRA were mixed in a solution with the concentrations of 40 mg/L of each antibiotic (TMP, SM, TET), and 300 mg/L of TRA altogether.

### 2.5. Pertraction Process

Pertraction experiments were performed in a closed, cylindrical stainless-steel cell with a 5.8 cm diameter and 6 cm length, as represented in Figure 2. The cell consisted of two compartments separated by a stainless membrane holder disc where the tested membranes were placed. Each membrane side compartment had a volume of 70 mL. The experimental temperature was kept at 25 °C for all pertraction tests using a thermostat. The tested membranes were roundly shaped with diameter of 2.5 cm. The active side of the membrane faced the feed solution. Both feed and permeate solutions were stirred with a magnetic stirrer during the whole experiment.

### 2.6. Diffusion Coefficient Calculation

Diffusion coefficients were evaluated using a simplified analytical model, based on Fick’s 1st law. This law describes mass transport through the membrane, where diffusion is the only transport mechanism through a homogenous medium, as follows: (1)$$\frac{1}{A}\frac{dn}{dt}=-D\frac{dc}{dx}$$
where n is the molar amount of the pharmaceutical that diffused through tested membrane, A is the active surface of the membrane, c is the pharmaceutical concentration, and D is diffusion coefficient of tested membrane material medium. Variables t and x are time and directed distance of diffusion process, respectively.

Considering the quasi-steady state of diffusion, where feed and permeate concentrations converge to an equilibrium and the adsorbed pollutant on the membrane is insignificant, also by assuming a constant volume for both feed and permeate solutions $\left(V_{f}=V_{p}=V\right)$ and a homogenous membrane material with a linear pharmaceutical concentration profile across the membrane thickness (𝛿), Equation (1) can be expressed as the following:

(a)
(2)$$\frac{dc_{p}}{dt}=\frac{DA}{V}\frac{\left(c_{f}-c_{p}\right)}{\delta }$$

(b)
(3)$$\frac{dc_{f}}{dt}=-\frac{DA}{V}\frac{\left(c_{f}-c_{p}\right)}{\delta }$$
where δ is the thickness of the studied membrane and c*_f_* and c*_p_* are the concentrations of the feed and permeate solutions, respectively.

We suppose THAT the amount of adsorbed pollutant on the membrane is negligible compared to the amount in both feed and permeate solutions, with the following given:(4)$$\left(V\times c_{p}\right)+\left(V\times c_{f}\right)=V\times c_{0}$$
where $c_{0}$ refers to the initial concentration of the feed solution. 

By merging Equation (1) with condition b and integrating the differential equations using initial experimental conditions [a) $c_{p}\left(t=0\right)=0\;$and b) $c_{f}\left(t=0\right)=c_{0}\;$], the relationship for calculating the diffusion coefficient is obtained following Equation (5), as follows:(5)$$D=-\frac{V\delta }{2At}\ln\frac{\left(c_{f}-c_{p}\right)}{c_{0}}$$

## 3. Results

We first present the membrane characterization results, followed by the pertraction results for tests performed on membranes with different model pollutants.

### 3.1. Membrane Characterization 

#### 3.1.1. Contact Angle

All four membrane contact angle measurements are gathered in Table 2. The contact angles shows surface hydrophobicity improvement of 10° noticed on both modified GMA-DMAEMA membranes. The contact angle MMA-DMAEMA 10 was 97° thanks to MMA hydrophobicity, and PDMD remained the most hydrophilic with a 50° contact angle. This hydrophobicity changes proves the successful grafting of the membranes. Moreover, the gathered results from this analysis contribute to explain differences between membranes’ interaction with pollutants.

#### 3.1.2. Scanning Electron Microscopy

Figure 3 groups the SEM images of all three modified PDMS surfaces from the top and one side-cut image of the GMA-DMAEMA 10 membrane. The side-cut SEM image (a) of modified membrane GMA-DMAEMA 10 shows three distinct layers—PET fibers, deposited PDMS, and grafted GMA-DMAEMA—barely visible on the top of the PDMS layer. However, top surfaces imaging shows a difference between the surfaces; on all three membranes, we can notice wrinkles and agglomerates due to membrane modification. The observed membranes’ surficial morphology is in accordance with the results obtained on contact angle tests, especially the top surface of (c) GMA-DMAEMA 10, which shows more superficial monomer crystallization, resulting from a higher surface hydrophobicity. The membrane surface imaging provides information on the membrane surface appearance, which will be an interpretation tool for the separation results.

#### 3.1.3. Fourier-Transform Infrared Spectroscopy

The FTIR analysis of PDMS and the modified membrane surfaces are compared in Figure 4. Using this analysis, it was important to visualize the functional groups of the modification molecules. For all three modified membranes, the shifted resonance of a conjugated ester function at 1725 cm^−1^ of the three modification molecules, methyl methacrylate, glycidyl methacrylate, and 2-(dimethylamino)ethyl methacrylate, were detected [26]. The C–N stretch of amine group of 2-(dimethylamino)ethyl methacrylate could be detected in the 1250–1020 cm^−1^ region [27]. However, the adjacent intense peaks at 873 cm^−1^, 1019 cm^−1^, and 1246 cm^−1^ corresponding to the Si–O stretch, O–Si–O symmetrical stretch, and CH_3_–Si/ CH_3_–Si–CH3, both deformation and scissoring stretches, respectively, eclipses the amine group bend [28,29,30]. Alternatively, DMEAMA groups may be totally covered; then, the amine function remains unobserved in the spectra of the modified membranes. The ester resonance detected at 1725 cm^−1^ proves the successful modification of the three membranes, and is in accordance with the previously published reports of grafted poly(methacrylate) polymers on membranes [31]. The observed absorbance at the region 3000–2769 cm^−1^ is linked to C–H stretch of the alkaline groups [28,32].

#### 3.1.4. Specific Surface Characterization

Both Table 3 and Figure 5 summarize specific surface characterization values and pore size distribution curves of the tested membranes. All four analyzed membranes showed a non-porous structure with a weak specific surface and low porosity structure. For example, the bare PDMS membrane specific surface is 12 m^2^/g and presents superficial macro and meso-pores (1 nm < meso-pores “r” < 25 nm < macro-pores “r” < 50 nm). Membrane grafting modification had an insignificant effect on MMA-DMAEMA 10 (d), and GMA-DMAEMA 5 (C) specific surfaces and even porosity. However, GMA-DMAEMA 10 (b) revealed a very week specific surface (S_BET_) with a value of 1.8 m^2^/g. The specific surface drop is due to superficial agglomerations formed during grafting. 

#### 3.1.5. Energy-Dispersive X-ray Spectroscopy

Element identification analysis of modified membrane surfaces are compared with the PDMS membrane composition from literature, listed in Table 4 [14]. The modified membrane analysis shows an increase in the carbon composition due to the structural carbon of the grafted molecules. In addition, nearly a total decline in the silicon ratio comparing modified to unmodified surfaces was noticed. This is due to the grafted molecules fully covering the PDMS surface. 

Moreover, nitrogen detection on a modified membrane proves the presence of amine groups from 2-(dimethylamino)ethyl methacrylate. The 20% difference in nitrogen percentage between GMA-DMAEMA 5 and both GMA-DMAEMA 10, and MMA-DMAEMA 10 mainly concerns UV grafting duration. Both MMA-DMAEMA 10 and GMA-DMAEMA 10 were exposed to 10 min of surface modification, allowing a larger amount of the grafted monomer (methyl methacrylate, or glycidyl methacrylate) to cover the 2-(dimethylamino)ethyl methacrylate, unlike the GMA-DMAEMA 5 membrane, which had lower grafting duration, leaving some 2-(dimethylamino)ethyl methacrylate groups uncovered and detectable during EDX analysis.

### 3.2. Pertractions

#### 3.2.1. Mixed and Separated Antibiotic Pertraction

Aiming to study membrane–pollutant interaction and pollutants’ synergic effects on separation process, pertraction using the PDMS membrane was performed with three different antibiotics (ATBs): SM, TMP, and TET model solutions (both mixed (MIX) and each pollutant separately) (see Figure 6).

During separated ATB pertraction, sulfamethoxazole was the best-transferred pollutant through the membrane, unlike tetracycline, which was highly adsorbed on the membrane but could not diffuse through it. This can be explained by the high interaction surface and adsorption affinity that TET presents compared to SM and TMP [33,34]. However, it was noted that in the MIX pertraction test, SM and TMP had similar permeation tendencies and TET could diffuse through the membrane and appear in the permeate side. TET permeation was exclusively noticed in the MIX pertraction test. Even though TET permeated through the nonporous membrane, it remained the most adsorbed antibiotic. Similar TET behaviors were noticed using highly porous modified carbon nanotube membranes [35]. We notice that TET dominates the adsorption on PDMS surface since the start of the experiment. Due to TET high interaction, it was even co-diffused during the permeation of SM and TMP throughout the non-porous membrane [33,34].

#### 3.2.2. Tramadol Pertraction

To highlight the membrane surface effect, the pertraction of highly concentrated tramadol as a model solution was tested using bare and grafted PDMS membranes. The UV-grafted membranes concern MMA-DMAEMA (10 min) and GMA-DMAEMA (5 and 10 min). The profile curves in Figure 7 show that tramadol’s adsorption was high during the first five days but decreased during the following days for all the membranes. Tramadol was detected in the permeate side of all tested membranes, which proves successful diffusion of tramadol through them. However, bare PDMS and the MMA-DMAEMA 5 grafted membrane reached a maximum permeation and remained constant, unlike both GMA-DMAEMA grafted membranes, which kept tramadol diffusion during all 30 days experiment. Different factors can be involved in membranes’ behavior towards TRA, such as membrane swelling or clogging, ionic forces, or even grafted monomers effect [15,18,36]. In further experiments, bare PDMS and both GMA-DMAEMA grafted membranes were used to investigate the synergic effect of tramadol and the three antibiotics during the pertraction process.

Figure 8 represents the diffusion coefficient calculation model, which corresponded perfectly to the experimental one of both GMA-DMAEMA 10 and GMA-DMAEMA 5 membranes. However, MMA-DMAEMA 10 and the PDMS membrane could match the mathematical model only during the early stages of the experiment. A negative deviation from the calculated model was noted on the 5th and 17th days of the experiment on PDMS and MMA-DMAEMA 10 membranes, respectively. The PDMS membrane with a bare surface showed the best diffusion coefficient, reaching 13.9 × 10^−12^ m^2^/s, followed by the modified GMA-DMAEMA 5 membrane with 4.54 × 10^−12^ m^2^/s. Even though these two membranes (PDMS and GMA-DMAEMA 5) have close specific surface area, total pore volume reduction affected tramadol diffusion process. The GMA-DMAEMA 10 membrane had the lowest diffusion coefficient of 4.008 × 10^−12^ m^2^/s. This is directly linked to a low specific surface, total pore volume, and formed crystals on the GMA-DMAEMA 10 membrane surface.

#### 3.2.3. Tramadol and Mixed Antibiotic Pertraction

For investigating all bare and GMA-DMAEMA grafted PDMS membranes’ behavior, the pertraction of the model solution loaded all with the tramadol and antibiotics mixtures (SM, TMP, and TET) was performed (see Figure 9). Concerning tramadol, high membrane adsorption was spotted during the first days of pertraction, mainly due to the higher concentration of tramadol and the diffusion process at its early stages. The permeate profile shows that GMA-DMAEMA 5 and bare PDMS membranes are much better for tramadol diffusion compared to GMA-DMAEMA 10, as proven previously by diffusion coefficients. Moreover, the GMA-DMAEMA 5 and bare PDMS membrane proved a positive linear regression, where the GMA-DMAEMA 5 curve had the highest regression coefficient. Regarding TMP, the bare PDMS membrane showed the best performance of adsorption and permeation compared to modified GMA-DMAEMA 5 and GMA-DMAEMA 10, which showed a nearly null permeation for TMP. The weak TMP permeation of the GMA-DMAEMA 5 membrane is mainly due to TMP permeation resistance of grafted monomers on the membrane. Similar behaviors were observed on the SM pollutant, where the bare PDMS membrane permeates SM much better than TMP. Comparable behaviors were noticed on GMA-DMAEMA 5 toward SM pollutants. However, GMA-DMAEMA10 remains impermeable for both TMP and SM, except for TRA, which could diffuse through it. Finally, the most adsorbed pollutant for all three membranes, bare PDMS, GMA-DMAEMA 5, and GMA-DMAEMA 10, is TET, which could not diffuse throughout any of the three tested membranes.

## 4. Discussion

Contact angle, SEM/EDX, FTIR, and surface measurements confirmed the surface modification of the three UV-grafted PDMS membranes. The increase in hydrophobicity supports the pollutant–membrane surface interaction. Once the pollutant adsorbs on the surface of the membrane, it diffuses to the permeate due to the concentration gradient driving force [37]. However, GMA-DMAEMA 10 modification decreased the specific surface of the membrane. This is explained by SEM imaging, proving the formation of crystals on the membrane surface during UV-grafting. The separation of antibiotics using the PDMS membrane for separated and mixed ATBs raised some remarks about membrane–ATB interactions. In both cases (separated, and mixed ATBs), TET was the most adsorbed on the PDMS membrane. Moreover, SM and TMP were the only ones that could permeate during the separated ATB pertraction test, exceptionally during the mixed ATBs pertraction, where TET was co-eluted and detected in the permeate side. The concentration of permeated TET was still minor compared to TMP and SM. This could be due to the co-elution of the pollutants through the membrane with other ATBs [33,34].

Testing the bare and modified membranes on tramadol revealed the different interactions with the grafted monomer. The adsorption of tramadol on all four tested membranes was observed in the early stages of the pertraction experiment. Even though all tested membranes permeated the tramadol, the MMA-DMAEMA 10 and the bare PDMS membrane reached maximum permeation and stopped diffusing TRA in the late stages of the separation; meanwhile, both GMA-DMAEMA 10 and GMA-DMAEMA 5 membranes kept positive regressions of TRA permeation. Those permeate changes are highlighted in the tramadol diffusion curves of each tested membrane. Outstandingly, tramadol’s diffusion factor on the bare PDMS membrane was much higher than all three modified membranes [15,18,36]. 

The last pertraction experiment, where all three ATBs and TRA mixtures were assessed on bare PDMS and GMA-DMAEMA (5 and 10) membranes, shows that both bare PDMS and the GMA-DMAEMA 5 membrane permeate TMP, SM, and similar amounts of TRA, but not TET, unlike GMA-DMAEMA 10 membrane behavior, which was selective during the permeation the pollutant. The GMA-DMAEMA 10 was passive to TMP, SM, and TET. However, concerning the TRA permeation, GMA-DMAEMA 10 was specifically selective, regardless of the drawback of slow permeation, which is blamed on the crystals formed during the UV-grafting of the membrane. Remarkable selectivity of GMA-DMAEMA 10 toward TRA may be explained with different results gathered during this study. Membrane hydrophobicity factor is to be excluded. GMA-DMAEMA 5 presented a slightly higher contact angle than GMA-DMAEMA 10, but GMA-DMAEMA 5 was unselective. Nevertheless, GMA-DMAEMA 5 grafted monomers have proven visible permeation resistance to SM and TMP, as reported in Figure 9. This partial permeation resistance is due to uncompleted GMA grafting on the PDMS membrane, which is shown in Table 4 by the presence of a 28.4% uncovered DMAEMA amine group on the GMA-DMAEMA 5 surface, unlike GMA-DMAEMA 10, showing specific selectivity to TRA. The membrane selectivity is related to a complete surface modification, affecting membrane specific surfaces, surficial pores, and membrane diffusion. Membrane selectivity can be implemented in different domains, such as sensor development or even tramadol recovery [38,39]. Further research should investigate and improve the flux and diffusion coefficients of selective membrane GMA-DMAEMA 10.5. 

## 5. Conclusions

The separation of ATBs using PDMS membranes has proven the possibility of co-elution of the pollutant during the separation process, specifically during the pertraction of MIX ATBs, where the TET was co-eluted due to SM and TMP permeation. Membrane modification impacted membrane diffusion ability, which was observed during the TRA separation experiment. However, the modified PDMS membrane GMA-DMAEMA 10 gained permeation selectivity toward TRA over different present ATBs. Further investigation is required to improve the feed and permeability of the selective membrane.

## Figures and Tables

**Figure 1 membranes-11-00752-f001:**
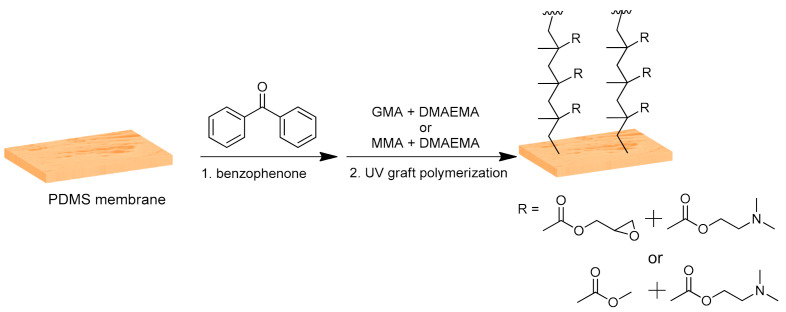
Ultra-violet grafting of polydimethylsiloxane membranes with the monomers glycidyl methacrylate or methyl methacrylate with 2-(dimethylamino)ethyl methacrylate.

**Figure 2 membranes-11-00752-f002:**
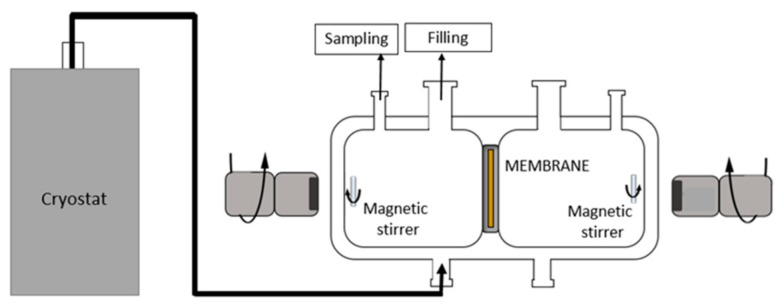
Schematization of the pertraction cell used for our experiments [25].

**Figure 3 membranes-11-00752-f003:**
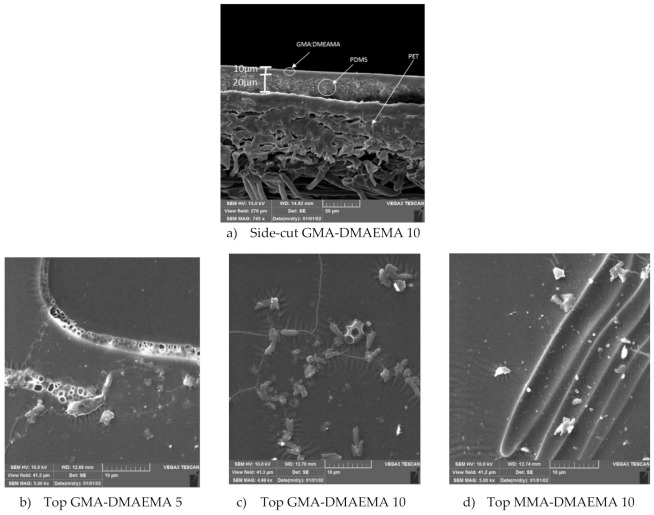
Side-cut and top surface Scanning Electron Microscopy imaging of modified GMA-DMAEMA and MMA-DMAEMA membranes.

**Figure 4 membranes-11-00752-f004:**
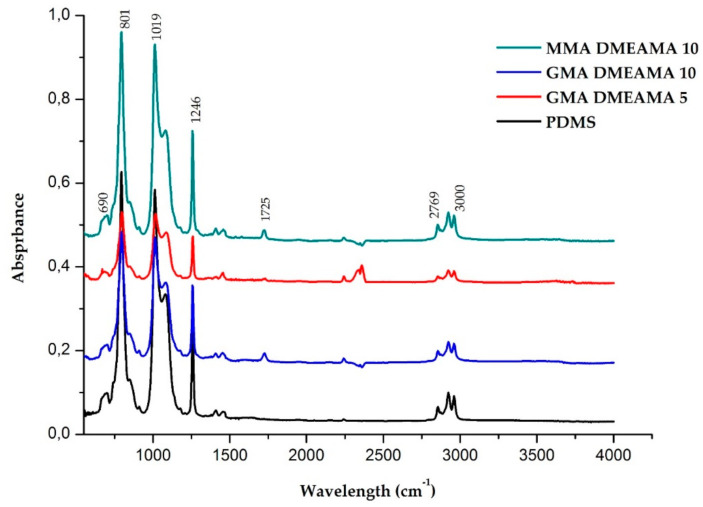
Fourier transform infrared spectra of bare and modified PDMS membranes on an active surface.

**Figure 5 membranes-11-00752-f005:**
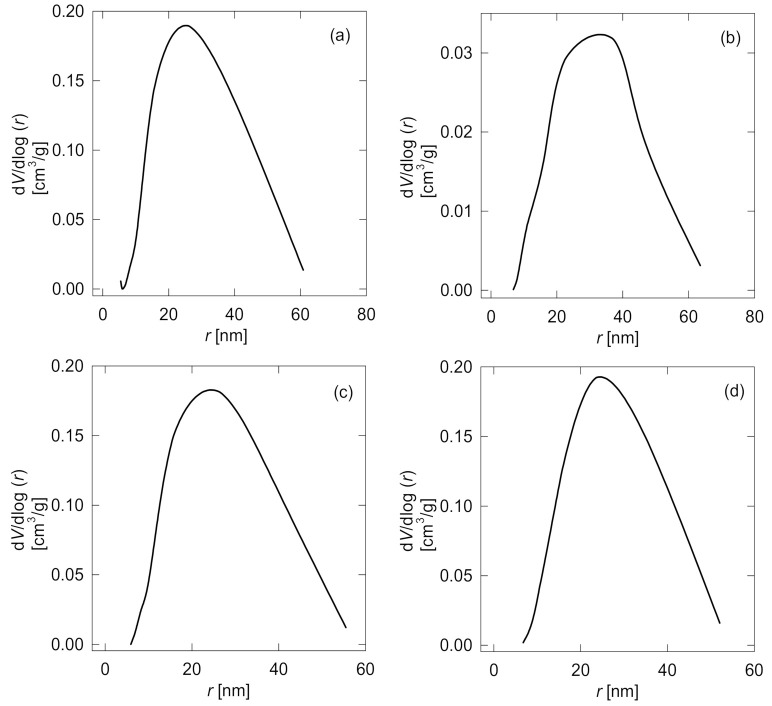
BJH curves for pore distribution visualization of (**a**) bare PDMS, (**b**) GMA-DMAEMA 10, (**c**) GMA-DMAEMA 5, and (**d**) MMA-DMAEMA 10 membranes.

**Figure 6 membranes-11-00752-f006:**
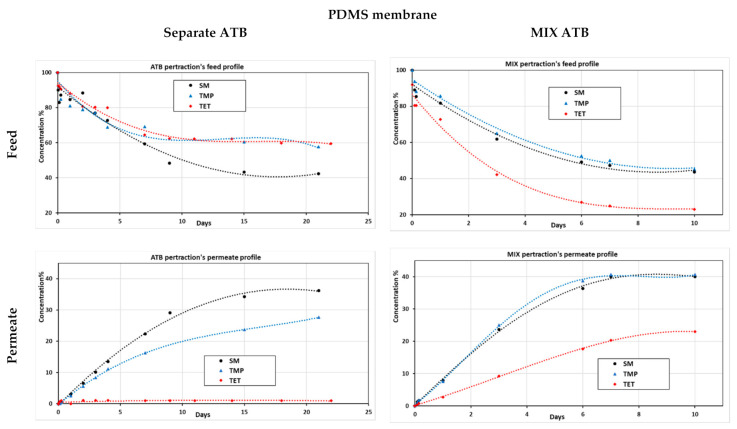
Adsorption, permeate, and feed profiles of both mixed and separated antibiotic solutions (sulfamethoxazole, trimethoprim, and tetracycline) of the pertraction process with a bare PDMS membrane.

**Figure 7 membranes-11-00752-f007:**
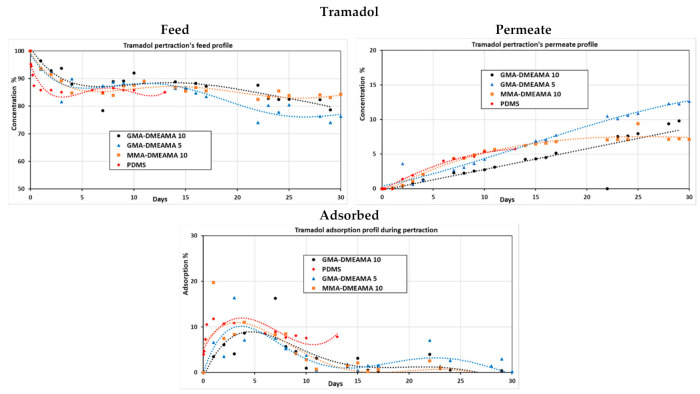
Adsorption, permeate, and feed profiles of the tramadol solution during the pertraction process with bare and modified PDMS membranes.

**Figure 8 membranes-11-00752-f008:**
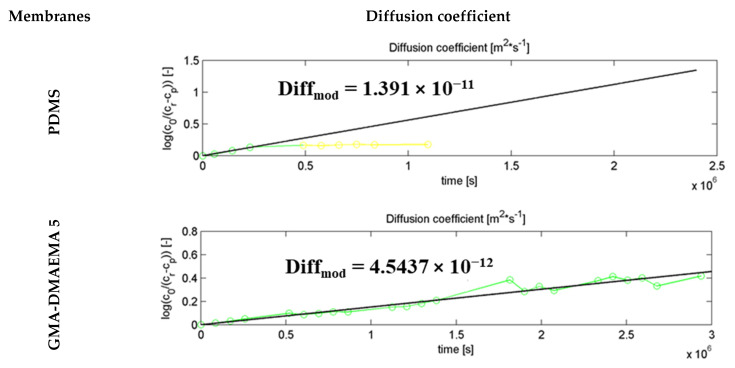
Experimental (linked point) and theoretical (line) diffusion coefficient of tramadol during the pertraction process with bare and modified PDMS membranes.

**Figure 9 membranes-11-00752-f009:**
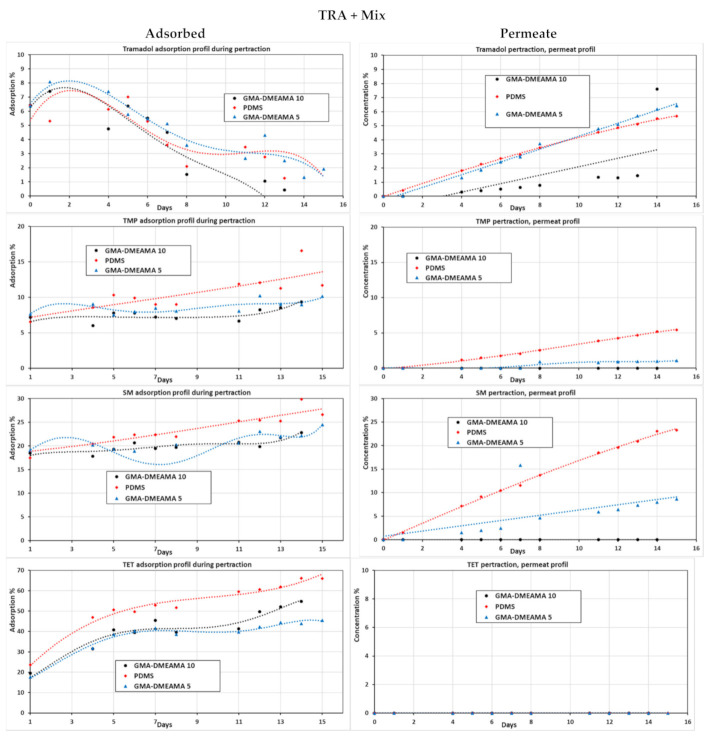
Adsorption and permeate profiles of tramadol, trimethoprim, sulfamethoxazole, and tetracycline together in the model solution of the pertraction process with bare and GMA-DMAEMA-modified PDMS membranes.

**Table 1 membranes-11-00752-t001:** Summary of the prepared solution for membrane tests.

Solutions	Separate ATB			MIX ATBs	TRA	ATBs + TRA
Pollutants	SM	TMP	TET			
SM (mg/L)	200	-	-	100	-	40
TMP (mg/L)	-	200	-	100	-	40
TET (mg/L)	-	-	200	100	-	40
TRA (mg/L)	-	-	-	-	500	300

**Table 2 membranes-11-00752-t002:** Ultra-pure water droplet contact angle results on the four tested membranes.

Membranes	Contact Angle (Degrees)
PDMS	50°
MMA-DMAEMA 10	97°
GMA-DMAEMA 10	57°
GMA-DMAEMA 5	63°

**Table 3 membranes-11-00752-t003:** Specific surface characterization results of all tested membranes.

Sample	S_BET_ (m^2^/g)	S_meso_ (m^2^/g)	V_tot_ (mm^3^liq/g)
PDMS	12	12	86
GMA-DMAEMA 10	1.8	1.6	12
GMA-DMAEMA 5	14	11	67
MMA-DMAEMA 10	10	10	67

S_BET_: specific surface; S_meso_: mesoporous surface; V_tot_: total porous volume.

**Table 4 membranes-11-00752-t004:** Energy-dispersive X-ray analysis of the elemental composition of modified membranes compared to the bare PDMS membrane composition from the literature [14].

EDX Analysis.	GMA-DMAEMA 10	MMA-DMAEMA 10	GMA-DMAEMA 5	PDMS [14]
Carbon (%)	71.46	67.75	60.65	47
Nitrogen (%)	4.55	6.09	28.45	0
Oxygen (%)	22.46	25.92	10.90	29
Silicon (%)	1.53	0.24	0	23

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
