# Peer review of "Glycidyl and Methyl Methacrylate UV-Grafted PDMS Membrane Modification toward Tramadol Membrane Selectivity"

_membranes, 2021, doi:10.3390/membranes11100752_

Round 1
Reviewer 1 Report
This manuscript needs extensive revision and upgrade.
(1) English needs to be modified and improved.
(2) Typo: Introduction part, line 58 “fulling”
(3) The sentence in lines 63-64 needs to be finalized.
(4) It is necessary to add more explanation about the reason why PDMS was selected as a research subject.
(5) It is easier to functionalize the membrane surface than grafting, so why grafting?
(6) Is there any physical damage on the membrane surface by UV treatment?
(7) Why contact angle of MMA-DMAEMA10 was increased to 97˚?
(8) Give a number to pictures in Figure 3.
(9) Modify the figure 4 graph to look good, and assign peak positions to all meaningful peaks on the graph. And, cite all the references which used to assign the peaks.
(10) Write all subscripts of the parameters in Table 3 correctly.
(11) Lack of detailed explanation and analysis: Table 2, Figure 3, Figure 4, Table 3, Figure 5, Table 4, Figure 6.
(12) Add all appropriate citations to the claims made based on previously published research results.
Reviewer 2 Report
The manuscript discusses a selective membrane by UV-grafting PDMS toward tramadol. Some problems should be addressed.
- Please provide the characterization of PDMS including pore size, etc. How does it take effect upon antibiotic pertraction ?
- In section 3.1.4, is the pore size distribution useful for understanding the selectivity? How does it affect the behavior?
- As for antibiotic pertraction tests, time is an important factor. How do you choose a suitable time?
- From Fig.8, the fitted diffusion coefficient of tramadol seems to not good especailly for PDMS and MMA-DMAEMA 10. Can you give some explanaiton?
- English writing needs to be more formal.
Round 2
Reviewer 1 Report
Thanks for your effort and response.
It seems enough to publish.
Cheers!